# A Mapless Local Path Planning Approach Using Deep Reinforcement Learning Framework

**DOI:** 10.3390/s23042036

**Published:** 2023-02-10

**Authors:** Yan Yin, Zhiyu Chen, Gang Liu, Jianwei Guo

**Affiliations:** School of Computer Science and Engineering, Changchun University of Technology, Changchun 130012, China

**Keywords:** D3QN, exploration-exploitation, turtlebot3, n-step, auxiliary reward functions, path planning

## Abstract

The key module for autonomous mobile robots is path planning and obstacle avoidance. Global path planning based on known maps has been effectively achieved. Local path planning in unknown dynamic environments is still very challenging due to the lack of detailed environmental information and unpredictability. This paper proposes an end-to-end local path planner n-step dueling double DQN with reward-based ϵ-greedy (RND3QN) based on a deep reinforcement learning framework, which acquires environmental data from LiDAR as input and uses a neural network to fit Q-values to output the corresponding discrete actions. The bias is reduced using n-step bootstrapping based on deep Q-network (DQN). The ϵ-greedy exploration-exploitation strategy is improved with the reward value as a measure of exploration, and an auxiliary reward function is introduced to increase the reward distribution of the sparse reward environment. Simulation experiments are conducted on the gazebo to test the algorithm’s effectiveness. The experimental data demonstrate that the average total reward value of RND3QN is higher than that of algorithms such as dueling double DQN (D3QN), and the success rates are increased by 174%, 65%, and 61% over D3QN on three stages, respectively. We experimented on the turtlebot3 waffle pi robot, and the strategies learned from the simulation can be effectively transferred to the real robot.

## 1. Introduction

With the gradual development and growth of artificial intelligence, mobile robots based on artificial intelligence have provided various conveniences to society while improving social productivity, among which autonomous mobile robots are widely concerned for their ability to complete tasks independently in a given environment. The key to autonomous mobile robots is the ability to navigate autonomously, and the basis of navigation is path planning. It means finding a safe path from the starting position to the target position without colliding with any obstacle. According to the degree of information about the environment, path planning can be divided into global path planning and local path planning (also called real-time planning). This paper aims to explore the problem of local path planning for robots in dynamic environments.

Different ways of path planning are needed in different environments, for example, a completely known environment usually uses global path planning, while a partially known environment or a completely unknown environment requires local path planning. The robot navigation completes its own localization based on simultaneous localization and mapping (SLAM) [1,2], and then plans the path to the target location by global path planning, the accuracy of which depends on the accuracy of environment acquisition. Global path planning can find the optimal solution, but it requires accurate information about the environment to be known in advance and its poor robustness to the noise of the environmental model. Local path planning detects the robot working environment by sensors to obtain information such as unknown and geometric properties of obstacles. With high robustness to environmental errors and noise, this method can provide real-time feedback and correction of the planning results. But the planning results may not be optimal due to the lack of global environmental information. An advantage of learning-based agent navigation over SLAM-based navigation is that it does not require high-precision sensors to construct environmental maps to accomplish the navigation task, thus making the navigation process simpler and reducing costs. Deep reinforcement learning path planning methods have been very successful in tasks such as robotics [3,4,5], image enhancement [6], and recommender systems [7]. However, there are many challenges in solving path planning problems using deep reinforcement learning, one of which is that many deep reinforcement learning algorithms suffer from overestimation bias. Secondly, in path planning tasks since there is only one target location in most cases, the rewards are sparsely distributed in the environment, which empirically slows down the learning speed significantly and even leads to non-convergence. Therefore, it’s difficult for the commonly used reward to function in this environment. Third, the strategy commonly explored and utilized by deep reinforcement learning algorithms is the ϵ-greedy strategy. It indicates that the agent will randomly choose the action with a small positive probability ϵ when making a decision, leaving a probability of 1−ϵ to choose the action with the highest action value. However, this strategy has many problems such as the agent must choose the action randomly with epsilon probability, which can lead the agent to reselect actions that are considered wrong. Therefore, a stochastic approach such as ϵ-greedy, may be ineffective in path planning tasks [8].

This work uses DQN [9] as a baseline. But it has several drawbacks, one of of which is the overestimation problem. The goal strategy is a greedy strategy that selects actions based on the value of all actions. And the goal strategy is chosen by the ϵ-greedy algorithm, which includes maximization operations. It maximizes on the estimated value, which can also be seen as implicitly estimating the maximum value, which is one of the two causes of overestimation (maximization). And the other cause of DQN algorithm overestimation is bootstrapping, since the DQN algorithm uses part of the estimated Q-value calculated by DQN when calculating the temporal-difference (TD) target (as in Equation (Equation 1)). The utilization of the part of the estimated Q-value calculated based on $y_{t}$ to update itself causes bootstrap. Also, backpropagation using stochastic gradient descent (as in Equation (2)) would further promote the overestimation of values. As in Figure 1.
(1)$$\begin{array}{cc} y_{t} & =r_{t}+\gamma \underset{a}{\max Q}\left(s_{t+1},a;w\right), \end{array}$$
(2)$$\begin{array}{cc} w & =w-\alpha \left(Q\left(s_{t},a_{t};w\right)-y_{t}\right)\frac{\partial Q(s_{t},a_{t};w)}{\partial w} \end{array}$$
where $y_{t}$ in Equation (Equation 1) is the TD target value at time step *t*, $r_{t}$ is the obtained reward, γ is the discount factor, $\underset{a}{\max Q}\left(s_{t+1},a;w\right)$ is the highest Q-value among all actions at next state $s_{t+1}$, $s_{t+1}$ is the state at time step *t*+1, *a* is the action at time step *t*, and *w* is the weight of the neural network. The α in Equation (2) is the learning rate, $Q(s_{t},a_{t};w)$ is estimated value, $y_{t}$ is true value. The network weights are updated by Equation (2), which makes the network converge toward the global optimal solution.

The maximization problem is further illustrated here. Let the observed true values be $x_{1},x_{2},x_{3},...,x_{n}$, and add noise with mean 0 to the true values to obtain $Q_{1},Q_{2},Q_{3},...,Q_{n}$. The zero-mean noise does not affect the mean: $E\left[mean_{i}\left(Q_{i}\right)\right]=mean_{i}\left(x_{i}\right)$. The zero-mean noise increases the maximum: $E\left[max_{i}\left(Q_{i}\right)\right]\geq max_{i}\left(x_{i}\right)$. In the DQN algorithm, set the maximum value of the action value function to *q*: q=maxQ(s,a;w). We get q⩾max(x(a)). Thus, the *q* is overestimated, and by the same time step *t*+1 $q_{t+1}$ is also higher than the true action value. Therefore, TD target is also an overestimate. TD learning drives action value function value $Q(s_{t},a_{t};w)$ towards $y_{t}$, leading it to be higher than the true value, which creates a vicious circle of overestimation.

The DQN algorithm idea is to output the value of each action through the network and select the highest of them to execute. If the overestimation phenomenon is uniform for all actions, it will not affect the highest-valued action to be selected. Because states and actions in the replay buffer occur at different frequencies, the DQN overestimation phenomenon is not uniformly generated, leading to the possibility that the action selected by DQN is non-optimal. The action value evaluation is not accurate enough. In the learning process, selecting different actions in some states may not have an impact on the Q-value, so we do not need to learn the impact of each action on that state. Decoupling the action-independent states from the calculation of the Q-value and using each action individually for the evaluation would result in more robust learning. The second challenge is the long training time. Reinforcement learning requires constant interaction with the environment for “trial and error”, so training times are usually long.


The RND3QN algorithm is proposed to improve some shortcomings of the current robot deep learning framework applied to autonomous navigation. It improves the model by the following points: shaping the auxiliary reward function; improving the exploration-exploitation strategy; introducing double DQN [10], dueling DQN [11], and using the n-step method guidance method. The above improvements effectively solve these problems and improve the effectiveness of local path planning.

The robot senses its surroundings through LIDAR. The process removes invalid data, sets null to 0 and infinity to 3.5, and then passes reasonable data (including observed environment information and its own location information) to the RND3QN algorithm to achieve local path planning in dynamic environments. The change of ϵ probability is controlled by the reward value, which balances the exploration or exploitation strategy. The task-appropriate ϵ decay formulation is also designed to avoid problems such as agents getting into local optimal solutions or overfitting.

We conducted simulation experiments and real experiments by using the robot operating system (ROS). As in Figure 2. The experiments proved that the total reward curves of double DQN [12], dueling DQN [11] and other algorithms have high volatility, and the total reward values of these algorithms present significantly lower than RND3QN. And RND3QN is more stable and has a significantly higher success rate. As a result, local path planning in an unknown dynamic environment is achieved. The main contributions are as follows:
For reward function shaping, a potential-based reward shaping is introduced. Two variables, angle and distance, are used as sources of potential energy. The robot is encouraged to move towards the target by continuously adjusting the angle and reducing the distance to the target.
Navigation tasks usually have only one goal and belong to a sparse reward environment. In this environment, the one-step method makes the agent learn slowly. Replacing the n-step method with a one-step method to obtain more information about the environment is a solution. Meanwhile, the effect of reward value and target value normalization on the stability of the algorithm is also explored.
To address the drawbacks of exploration in traditional reinforcement learning, an ϵ-greedy strategy based on reward value control ϵ probability variation is proposed. It can make the agent use exploration rationally in the environment and learn more effectively.


The rest of the paper is organized as follows. Section 2 provides a summary of existing robot path planning algorithms. Section 3 first introduces the Markov decision process and the foundations of reinforcement learning. This is followed by a description of the problem definition and the RND3QN algorithm. Experimental results and a brief analysis are listed in Section 4. Section 5 summarizes the entire article and discusses future research that can be conducted in this area.

## 2. Related Work

This section describes the related work about path planning methods. To date, many methods have emerged for the path planning problem. We classify them into two main categories, traditional algorithms based on non-learning and learning, where the mainstream algorithms are based on deep reinforcement learning methods. The former mostly requires complete map information to complete path planning and cannot cope with unpredictable situations, while the latter can learn based on the data continuously collected by sensors without complete map information.

### 2.1. Traditional Algorithms

Traditional algorithms can be divided into two categories: graph search-based methods and sampling-based methods. The earliest graph search algorithm is Dijkstra’s algorithm proposed by dutch scientist E.W. Dijkstra in 1959 [13], which uses a greedy strategy, traversing the shortest and unvisited points of an edge at a time until the endpoint. Subsequently, the well-known A* algorithm [14] was proposed. This algorithm adds heuristic functions to Dijkstra’s algorithm to improve performance and accuracy. Firstly A* algorithm has poor real-time performance. Secondly, as the number of nodes increases, its computation and time grow exponentially. It also cannot handle dynamic obstacles. The D* algorithm [15], proposed by Anthony Stentz et al. in 1997, is capable of handling the planning problem of dynamic obstacles. D* reverses the direction of planning from the target point to the starting point, and can handle some partially or completely unknown environments and dynamic obstacle situations, solving the A* algorithm cannot deal with the dilemma of dynamic obstacles. The D* algorithm consumes more resources and has a slow computation time. Koenig et al. extended the D* and lifelong planning A* (LPA*) [16] algorithms and proposed the D* Lite algorithm [17] using reverse search with incremental search makes it possible to continuously update the optimal path from the current point to the target point in the dynamic obstacle map using the node distance information generated in the previous iterations, which reduces the algorithmic complexity of D* path planning and solves the LPA* iterative planning problem. Another class of research is sampling-based algorithms, and the research that laid the foundation is the rapidly-exploring random trees (RRT) algorithm proposed by LaValle et al. in 1998. The RRT algorithm [18] is a fast search algorithm at the expense of optimal behavior, and the path it searches is often not the optimal path. To solve this problem, RRT* [19] appears, which makes RRT possess asymptotic optimality, but this approximation to the optimum is not fast. To improve the convergence speed of RRT*, Informed-RRT* [20] is developed to improve the convergence speed of the initial path to the optimal path. Conventional path planning has its limitations that the robot can’t complete planning in complex unknown environments and quickly adapt to its surroundings because of its dependence on the environment.

### 2.2. Reinforcement Learning Algorithms

Markov Decision Process (MDP) [21] is a mathematical representation of the reinforcement learning problem. And path planning is a natural markovian decision process. The earliest research in value learning problem is solved by Q-learning [22]. However, it has disadvantages such as poor generalization ability, slow convergence and not being applicable to high-dimensional environments. In 2015, to solve the memory overflow problem caused by large amount of data or coming continuous actions in high-dimensional environments, DeepMind proposed the DQN [9] algorithm in Nature, which combines neural networks with Q-learning.

Neural networks are used to generate Q-values for actions, turning the Q table update problem into a function fitting problem. To address the problem of slow convergence of the DQN algorithm, the work of Jiang et al. proposes a combination of deep neural networks with empirical playback and prior knowledge [23] to speed up the convergence rate. Subsequently the work of Hasselt et al. proposes double Q-learning [10] for the overestimation problem of Q-learning, which alleviates the overestimation phenomenon. The neural network of the DQN algorithm is inaccurate in the way it calculates Q-value. Ziyu Wang et al. [11] made a simple network structure of the DQN improvement by splitting the network into two estimators, one for the state value function and the other for the action dominance function.

Distinguishing the state values that are not related to the action improves the accuracy of Q-value. The work of double Q-learning does not eliminate overestimation, and the average DQN [24] proposed by Oron Anschel et al. in 2017 demonstrates the existence of approximation error and the impact of overestimation on learning. The method sets up k networks and then takes the average of the previous Q-value estimates of these networks as the current Q-value, thus achieving a more stable training process and improving performance by reducing the variance of the approximation error in the target value. Benjamin Riviere et al. [25] propose distinguishable safety modules to ensure collision-free operation, and in environments, with many obstacles, the collision avoidance success rate improves by 20%. However, known environmental information is still required for planning. To address the problem of reward sparsity in path planning tasks, a 2018 state-of-the-art approach is random network distillation (RND) [26], a flexible combination of intrinsic and extrinsic rewards. In 2019 Haobin Shi et al. introduced an exploration strategy with intrinsic incentive rewards in [3], where curiosity is used to encourage an agent to explore unvisited environmental states and as an additional reward for exploratory behavior. The algorithm uses laser ranging results as inputs and outputs continuous execution actions. Experimental results show that the algorithm outperformed RND. Subsequent work [27] proposes a novel generative adversarial exploration approach (GAEX) that outputs intrinsic rewards to encourage exploration by an agent through generative adversarial networks that identify infrequently visited mounted states. To solve the problem that conventional algorithms need to re-invoke the planning algorithm to find an alternative path every time the robot encounters a conflict. A global guided reinforcement learning approach (G2RL) [28] is proposed. The method adopts a double DQN based on prioritized experience replay in local path planning and an A* algorithm in global path planning. It also incorporates a new reward structure to make it more general and robust. The network of the DQN algorithm has neither memory of prior observations nor long-term prediction capability, so Koppaka Ganesh Sai Apuroop et al. [29] combined deep reinforcement learning algorithms with LSTM [30] to solve the path planning problem of cleaning robots indoors, allowing the robot to generate a least costly path in a shorter time. In the latest research on catastrophic forgetting in artificial neural networks, Qingfeng Lan et al. in [31] reduced forgetting and maintained high sample efficiency by integrating knowledge from the target Q-network to the current Q-network, while weighing the strategy of learning new knowledge and preserving old knowledge. Replay buffer may have a large amount of duplicate and invalid data, leading to a waste of memory space. It also reduces data utilization. There has been related work on improving the replay buffer, such as that proposed by Tom Schaul et al. in 2015 prioritized experience replay (PER) [32] uses prioritized empirical replay to replace the sampling method with uniform sampling to non-uniform sampling. Each piece of data has a different priority, and the most useful information is learned first, thus improving data utilization. Subsequent work [33] proposed a novel technique hindsight experience replay(HER) that combines the characteristics of sequential decision problems in reinforcement learning HER enables agent to learn from failed experiences, solving the problem of reward sparsity in reinforcement learning. In 2019 Rui Zhao et al. proposed a simple and efficient energy-based method [34] to prioritize playback of “posterior experience”, innovatively using “trace energy” instead of TD-error as a measure of priority. In tasks such as continuous control [35], an actor-critic reinforcement learning algorithm is used to achieve autonomous navigation and obstacle avoidance of UAVs, and robustness to unknown environments through localization noise. Junjie Zeng et al. proposed the MK-A3C [36] algorithm for memory neural networks to solve the problem of continuous control navigation of a robot in an unknown dynamic environment for incomplete robots, and trained using migration learning.

## 3. Materials and Methods

### 3.1. Background

The path planning is constructed as a partially observable markov decision process (POMDP) [37] model. At each discrete moment t=0,1,2,3,..., both the agent and the environment interact, and the agent chooses an action $a_{t}\in A\left(S\right)$ based on observing some feature expression $s_{t}\in S$. At the next moment, because of taking the action, a numerical reward, $r_{t+1}\in R$, is given, and then the agent enters a new state. Thus, the agent and the environment jointly create a trajectory $s_{0},a_{0},r_{1},s_{1},a_{1},r_{2},s_{2},a_{2},r_{3},...$.

The goal of agent is formally characterized as a special signal, namely reward, which is transmitted to the agent through the environment, and at each moment, the reward is a single scalar value, $r_{t+1}\in R$. The reward return represents the sum of future rewards $G=r_{t}+r_{t+1}+r_{t+2}+r_{t+3}...$. As the network gradually converges, the accuracy of the agent’s predictions about the future decreases. Therefore ‘discount’ is applied to future rewards. The cumulative discounted reward $U_{t}=r_{t}+\gamma r_{t+1}+\gamma r_{t+2}+\gamma r_{t+3}...$ represents the sum of the cumulative discounted future rewards, where γ is a parameter, 0≤γ≤1, which is called the discounted factor. When γ=0, the agent only cares about the gain of the current state, and as γ increases, the discounted return will consider more about the future gain. The purpose of reinforcement learning is to maximize discounted return. The return of adjacent time steps can be related to each other by Equation (Equation 3).
(3)$$\begin{array}{cc} U_{t} & =R_{t}+\gamma r_{t+1}+\gamma r_{t+2}+\gamma r_{t+3}...\\ \; & =r_{t+1}+\gamma \left(r_{t+2}+\gamma r_{t+3}+\gamma^{2}r_{t+4}\right)...\\ \; & =r_{t+1}+\gamma U_{t+1} \end{array}$$

A policy in reinforcement learning is a mapping between the selection probabilities of a state taking each action, and the formula for the policy π is π(a|s)=P(A=a|S=s). The value of taking action *a* at state s under strategy π is denoted as $q_{\pi }\left(s,a\right)$, which is called the action value function. The expected payoff of all possible decision sequences under strategy π starting from state *s* and after performing action *a*.
(4)$$\begin{array}{cc} Q_{\pi }\left(s\right)=E\left[G_{t}|S_{t}=s,A_{t}=a\right] & =E_{\pi }\left[\sum_{i=0}^{\infty }\gamma^{i}r_{t+i+1}|S_{t}=s,A_{t}=a\right] \end{array}$$

Because actions and states are random so the value of $q_{\pi }$ is random, and we call its maximum value the optimal action value function.
(5)$$\begin{array}{c} Q^{*}\left(s_{t},a_{t}\right)=\max_{\pi }Q_{\pi }\left(s,a\right) \end{array}$$

The agent is controlled by two functions π(a|s) and $Q^{*}\left(s_{t},a_{t}\right)$, and this work uses the optimal action value function to control the agent. The function is fitted by a neural network, using the DQN algorithm, which is the pioneer of deep reinforcement learning, as the baseline its target value is calculated as Equation (Equation 1), with a loss function $L={\left(target-Q\left(s,a;\theta \right)\right)}^{2}$.

### 3.2. Problem Definition

The robot is in an unknown environment and uses radar detectors to collect data to grasp information about the environment. The robot and the obstacles are three-dimensional objects in the real environment. To simplify the problem, this study ignores the height of the robot and the obstacles and treats them as two-dimensional objects. Without knowing the location, shape and size of the obstacles in the unknown environment, turtlebot3 only knows the start position and the target position.

Local path planning is to enable the agent to perform path planning and obstacle avoidance based on a partially observable environment, where the robot does not have access to all environmental information in an unknown environment and therefore cannot be constructed as a general MDP. This task requires a POMDP model [37].

Defining it as the tuple $M=\left\{S,A,T,R,\Omega ,O,\gamma \right\}$. The POMDP has state space S, action space A, transition function *T*, reward signal *R*, a discount factor γ∈ [0,1), a finite set of observations Ω and an observation space *O*. $\mathrm{Goal}=\left\{g_{1},g_{2},...,g_{n}\right\}$ represents a set of target position coordinate points. $\left(obs_{x_{j}},obs_{y_{j}}\right)$, (j=1,...,n) denotes the coordinate positions of n obstacles. The ambient state space is represented by S, which is the data obtained by the robot from the target detection and analysis by LiDAR (LDS). $S=LDS\left(24\right)+Distance\left(1\right)+Angle\left(1\right)+Obstacle_{angle}\left(1\right)+Obstacle_{min\_range}\left(1\right)$. We consider setting the action space as discrete, and the sizes of the action space and state space as in Table 1. The correspondence between action and velocity is shown in Table 2. The set $A=\left\{0,1,2,3,4\right\}$ represents five different actions in the action space. Meanwhile, we control the line speed at 0.25 m/s. The obstacles include static and dynamic obstacles.

### 3.3. The n-Step Dueling Double DQN with Reward-Based ϵ-Greedy (RND3QN)

We propose RND3QN, which enables the robot to perceive the environment and perform feature extraction through neural networks. When interacting with the environment, the data scanned by LiDAR is stored in the replay buffer, from which a batch of data of mini_batch size is randomly sampled as input. Since the n-step method is used, each data is n transitions, where n is used as a hyperparameter. This batch of data is processed through the evaluation neural network and fitted to a linear function that replaces the Q-table in the Q-learning algorithm, while periodically updating the target network. The DQN algorithm uses the ϵ-greedy algorithm to select actions. A drawback of this method is that it requires constant parameter tuning even in simple environments. Reward-based ϵ-Greedy is used in RND3QN. It inferred the current exploration-exploitation state from the reward signal and dynamically adjusted the change of ϵ according to this state to improve the exploration efficiency. We experimented with whether the soft update method used by deep deterministic policy gradient (DDPG) [38] would have a facilitating effect on the convergence of the network and found that the effect is slightly worse than being updated, and we need to adjust the weight hyperparameters manually, so the target network is updated using the hard update method. We summarize the agent and environment interaction model as shown in Figure 3.

#### 3.3.1. The n-Step TD

The n-step method allows flexible control of the backward sampling step size *n* while reducing errors. The one-step TD can only sample one transition with reward at a time, which is called one-step reward $G_{t}=r_{t+1}+\gamma Q\left(s_{t+1}\right)$. And n-step TD can sample *n* transitions with rewards and obtain more observations than one step, which is called multi-step return $G_{t:t+n}=r_{t+1}+\gamma r_{t+2}+...+\gamma^{n-1}r_{t+n}+\gamma^{n}Q_{t+n-1}$. Thus, n-step TD targets will have better learning efficiency. The n-step payoffs use the value function $V_{t+n-1}$ to correct the sum of all remaining rewards after $r_{t+n}$. The n-step payoffs have error-reducing properties, i.e., its expected worst error is guaranteed to be no larger than $\gamma^{n}$ times the worst error of $Q_{t+n-1}$.
(6)$$\begin{array}{c} \max\left|E_{\pi }\left[G_{t:t+n}|s_{t}=S\right]-v_{\pi }\right|⩽\gamma^{n}\max\left|V_{t+n-1}-v_{\pi }\left(s\right)\right| \end{array}$$
where $G_{t:t+n}$ denotes the return reward at time *t* to *n* steps.

Since the learning speed of the DQN algorithm is slow, we introduce the n-step bootstrap method into double dueling DQN and propose a new framework that can accelerate the convergence of the Q-network. The n-step DQN (ND3QN) method is centered on the expansion of the bellman equation, and since its formula is constantly iterative, we can derive it as Equation (Equation 7):(7)$$\begin{array}{cc} u_{t} & =r_{t}+\gamma u_{t+1}\\ \; & =r_{t}+\gamma \left(r_{t+1}+\gamma u_{t+2}\right)\\ \; & =r_{t}+\gamma r_{t+1}+\gamma^{2}r_{t+2}+\gamma^{2}u_{t+3}\\ \; & =\sum_{i=0}^{n-1}\gamma^{i}r_{t+i}+\gamma^{n}u_{t+n} \end{array}$$
where $u_{t}$ is the cumulative discount reward, *R* is the immediate reward, and γ is the discount factor. The n-step TD targets are used in the DQN algorithm, and the target value is calculated as follows:(8)$$\begin{array}{c} y_{t}=\sum_{i=0}^{n-1}\gamma^{i}r_{t+i}+\gamma^{n}\max_{a}Q^{*}\left(S_{t+n},a;\theta \right) \end{array}$$
where $y_{t}$ is the target value and θ is a parameter of the neural network. The n-step method payoffs use the value function $V_{t+n-1}$ to correct the sum of all remaining rewards after $R_{t+n}$.

#### 3.3.2. Mitigating Overestimation Using Double DQN

Double DQN is added to our algorithm to mitigate the overestimation. The selection of the optimal action and the calculation of the optimal action value function are coupled in the traditional DQN method, but this method decouples them to achieve the effect of alleviating overestimation. The action with the largest Q-value is obtained using the evaluation network parameters, and then the action is used for the calculation of the target so that the value of the target is smaller than the target value obtained by maximizing the Q-value. That is, the action is selected using the evaluation network, $a^{*}=\underset{a}{\arg\max}Q\left(S_{t+1},a;w\right)$. Target value is calculated using the target network. Thus the Q-value calculated by double DQN is less than or equal to the Q-value maximized by DQN, $Q\left(S_{t+1},a^{*};w^{-}\right)⩽\max_{a}Q\left(S_{t+1},a;w\right)$. In addition to this, we came up with the normalized value approach to further mitigate the overestimation problem, i.e., scaling the mean to normalize the reward function value and normalizing the target value of the value function in the same way. The standard deviation is obtained from the Q-value data generated by the previous DQN, and the target Q-value is divided by this standard deviation to normalize it to reduce the error and overestimation problem. However, the results were not satisfactory in the experiments, so the method was not used in this work.

#### 3.3.3. Introduction of Dueling Network to Optimize the Network

The Dueling network divides the Q-value approximated by the neural network into two quantities: the value of the state V(s), and the advantage of the action in this state $A\left(s,a\right)$. The advantage function is defined as Equation (Equation 9).
(9)$$\begin{array}{c} A^{*}\left(s,a\right)=Q^{*}\left(s,a\right)-V^{*}\left(s\right) \end{array}$$
where $A^{*}\left(s,a\right)$ indicates the advantage of action relative to baseline. If the advantage *A** is greater than 0, it means that the action is better than the average action. Conversely, it is worse than the average action. $Q^{*}\left(s,a\right)$ evaluates the goodness of action a made in state *s* and $V^{*}\left(s\right)$ evaluates the goodness of state *s* as the baseline.
(10)$$\begin{array}{c} V^{*}\left(s\right)=\max_{a}Q^{*}\left(s,a\right) \end{array}$$

Taking both sides of Equation (Equation 9) to be maximized the joint Equation (Equation 10) further yields that
(11)$$\begin{array}{cc} \max_{a}A^{*}\left(s,a\right) & =\max_{a}Q^{*}\left(s,a\right)-V^{*}\left(s\right)\\ \; & =0 \end{array}$$

From Equation (Equation 9), we get $Q^{*}\left(s,a\right)=A^{*}\left(s,a\right)+V^{*}\left(s\right)$, but the equation cannot uniquely determine *V** and *A** by *Q** (uniqueness), which will lead to not detecting the fluctuation of neural network *V* and *A* during training, thus reducing the accuracy of the algorithm. We then combine Equation (Equation 12) with $\max_{a}A^{*}\left(s,a\right)$ to resolve the uniqueness, and arrive at
(12)$$\begin{array}{c} \max_{a}Q^{*}\left(s,a\right)=V^{*}\left(s\right)+A^{*}\left(s,a\right)-\max_{a}A^{*}\left(s,a\right) \end{array}$$

In the experiment, the average was used instead of the maximum operation to obtain better results. The mathematical form of the dueling network is given below:(13)$$\begin{array}{c} Q\left(s,a;\theta \right)=V\left(s;\theta^{V}\right)+\left(A\left(s,a;\theta^{A}\right)-\frac{\sum_{a}A\left(s,a;\theta^{A}\right)}{|A|}\right) \end{array}$$
where *Q* is the action value function; *V* is the value function; and *A* is the dominance function. $\theta^{V}$ is the weight parameter of the state network, $\theta^{A}$ is the weight parameter of the advantage network *A*. Our network structure uses the dueling network, as in Figure 4, and outputs the action space size vector value and a real advantage value *A* through three fully connected layers, respectively. $V\left(s;\theta^{V}\right)$ is the state function, outputting a scalar, $A\left(s,a;\theta^{A}\right)$ is the advantage function, outputting a vector, and the vector length is equal to the action space size.

#### 3.3.4. Reward-Based ϵ-Greedy

We propose a reward-based ϵ-greedy approach to solve this decaying strategy problem. When an agent achieves a larger reward, it has a higher probability of choosing the optimal strategy. And when the reward obtained in the current round is less than or equal to the reward in the previous round, the agent will have a higher probability to choose the action randomly. This approach removes much of the uncertainty surrounding the nature of exploration by an agent, i.e., should the agent explore more? It is natural to think that this parameter controlling the degree of exploration is the reward, which is adjusted to obtain the result we want. This parameter depends heavily on the environment and the task. So it must be set according to the specific problem. RND3QN uses the reward to determine the ϵ decay, and the ϵ decreases only when the agent exceeds a certain reward threshold, otherwise the ϵ remains unchanged. Thus, the Q-value is close to the effort close to the optimal value. Also, each time the ϵ decreases, we set a higher goal for the agent and wait for the agent to reach the new goal. The same steps are repeated. The epsilon probability decay is calculated as in Equation (Equation 14) where the initial ϵ and the minimum ϵ are used as hyperparameters.
(14)$$\begin{array}{c} \epsilon =\left(\epsilon_{init}-\epsilon_{min}\right)\cdot \max\left(\frac{N-n_{step}}{N},0\right)+\epsilon_{min} \end{array}$$

Based on the above sections we propose an extended version of dueling double DQN by adding the n-step method and simply improving the ϵ-greedy exploration strategy. Where the pseudocode of the reward-based ϵ-greedy strategy is shown in Algorithm 1, and the pseudocode of the ND3QN as algorithm is shown in Algorithm 2.
**Algorithm 1** reward-based ϵ-greedy**Input**:
 Initialize total training episodes E, total steps G, ϵ-greedy initial probability ϵ, min ϵ$\epsilon_{min}$
 Initialize reward threshold R and reward increment I**Output**: selected actions
 **for** e = 1 to E **do**
  **if** epsilon > $\epsilon_{min}$ and $last_{reward}⩾reward_{threshold}$ **then**
   $\epsilon =\left(\epsilon_{init}-\epsilon_{min}\right)\max\left(\frac{E-G}{E},0\right)$
   R=R+I
  **end if**
  random selection of actions with the probability of ϵ
  select the optimal action with a probability of 1−ϵ
 **end for**

**Algorithm 2** ND3QN**Input**:
Initialize a dueling network Q(s,a;θ) and a target dueling network $Q(s,a;\theta^{-})$
   Initialize replay memory R, batch size B, discount factor γ, learning rate α, total training    episodes E, the max steps of each episode M
   Initialize n-step *n*, ϵ-greedy initial probability ϵ

**Output**: Q value for each action
 **for** i = 1 to B **do**
   Taking a random subscript finish in the interval (n, R.size)
   begin = finish − n
   Randomly take state and action in the interval (begin, finish)
   **for** t = 1 to n **do**
    $G\leftarrow \sum_{i=begin}^{min(begin+n,R.size)}\gamma^{\left(i-begin\right)}R_{i}$
  **end for**
 **end for**
 **for** e = 1 to episodes **do**
  **for** t = 1 to S **do**
    action *a* is chosen according to a reward-based ϵ-greedy policy
    get the next state, reward, and done from the environment
    store transition $\left(s_{t},a_{t},r_{t},s_{t+1}\right)$ in replay buffer
    **if** memory.length⩾train_start **then**
     randomly sample n pieces of transitions from the replay memory;
     obtain the optimal action
     $a^{*}=\underset{a}{\mathrm{argmax}}Q\left(S_{t+n},a;\theta \right)$            
$target=\left\{\begin{array}{ll} G & \mathrm{if}\;\mathrm{done}\\ G+\gamma^{n}Q\left(S_{t+n},a^{*};w^{-}\right) & \mathrm{else} \end{array}\right.$
    Calculate the loss $Q\left(s_{i},a_{i};\theta_{t}\right)-y_{i}{)}^{2}$ with network parameters θ
    update target network $\theta^{-}=\theta$ every C step
   **end if**
  **end for**
 **end for**


#### 3.3.5. Design of the Reward Function

The reward function is an important part of deep reinforcement learning, which determines the speed and degree of convergence of reinforcement learning algorithms. Reward shaping related work [39,40] has been studied to address the problem that agent does not recognize key actions and is not motivated to explore in more complex scenarios. Sparse rewards can lead to meaningful rewards that are not available most of the time during training, and it is difficult for the agent to learn in the direction of the goal without feedback. In contrast, an effective reward function not only accelerates the training speed of the algorithm but also improves the convergence accuracy of the algorithm. In this paper, we shape the auxiliary reward function and divide the reward function into two parts. The main line reward is designed as follows (1) collision obstacle penalty, when the robot hits an obstacle or hits a wall, it is given a large negative reward $r_{obstacle}$. (2) if the agent reaches the target point without collision, it is given a large positive reward $r_{goal}$.
(15)$$\begin{array}{cc} \; & Reward=\left\{\begin{array}{cc} r_{goal} & ,\;\mathrm{if}\;\mathrm{goal}\\ r_{obstacle} & ,\;\mathrm{if}\;\mathrm{obstacle} \end{array}\right. \end{array}$$

The sub-target of the auxiliary reward function reaction task echoes the state space and is designed as follows: the angle θ between the robot direction and the target, and the robot rotation angle is the yaw angle. The clockwise rotation angle is [0,2π]. When the angle is π or −π, the robot direction is exactly opposite to the target direction.
(16)$$\begin{array}{c} \left\{\begin{array}{cc} r_{\theta }⩾0 & ,\;\mathrm{if}\;-\frac{\pi }{2}<\theta <\frac{\pi }{2}\\ r_{\theta }<0 & ,\;\mathrm{otherwise} \end{array}\right. \end{array}$$

(1) A shaping reward is added to the original reward function. It is a potential energy function whose difference in distance from the final target determines the magnitude of the potential energy. That is, the closer the distance to the target, the higher the reward, and vice versa. The distance is calculated using the two-dimensional Euclidean formula. Euclidean distance is a commonly used definition of distance, which represents the true distance between two points in m-dimensional space, and this question discusses the problem in two dimensions, the equation of two-dimensional space is $d=\sqrt{{\left(x_{2}-x_{1}\right)}^{2}+{\left(y_{2}-y_{1}\right)}^{2}}$. The auxiliary reward from distance is calculated by the following equation
(17)$$\begin{array}{c} \left\{\begin{array}{cc} r_{d}>2 & ,\;\mathrm{if}\;D_{c}<D_{g}\\ 1<r_{d}⩾2 & ,\;\mathrm{otherwise} \end{array}\right. \end{array}$$
where $D_{c}$ denotes the current distance of the robot from the target, and $D_{g}$ is the absolute distance (the distance of the robot’s initial position from the target point), as shown in Figure Figure 5.
(18)$$\begin{array}{c} r_{d}=2^{\frac{D_{g}}{D_{c}}} \end{array}$$
where $R_{d}$ is the prize for distance. When the distance is smaller, it means that the robot is closer to the target point and thus receives more reward.

(2) Setting the source of the exceptional auxiliary reward as an angle to motivate the robot to orient itself with less possibility of taking a detour, and continuously guides the robot to the target position through successive rewards, where the angle reward is calculated by the Equation (Equation 20).
(19)$$\begin{array}{c} \varphi_{\theta }=4\left|\frac{1}{2}-\left\lfloor \left(\frac{1}{4}+\frac{\theta }{2}\;\mathrm{mod}\;2\pi \right)\right\rfloor \right| \end{array}$$
(20)$$\begin{array}{c} r_{\theta }=5\left(1-\varphi_{\theta }\right) \end{array}$$
where ‘⌊⌋’ mathematically means taking the integer part of the floating point number, the robot head, and the target point angle $heading=goal_{angle}-\phi$. The ϕ is the yaw angle, the yaw angle rotates around the y-axis, also called the yaw angle, the robot head yawing to the right is positive and vice versa, θ is the angle with the target, and $r_{\theta }$ is the reward from the angle. In the ros-based experiments, we obtained the yaw angle by obtaining the quaternion issued by odom, then converting it to an Euler angle and taking the heading angle of it.

As shown in Figure 6, the robot takes different actions corresponding to different heading angles, thus obtaining different reward values. The setting of the reward function affects the speed and degree of convergence of the algorithm. The reward function is divided into three parts main line reward and two parts of auxiliary reward, and the essential purpose is to let the robot reach the destination in the shortest time without collision. More than 500 steps per episode is a timeout and the turn ends. In contrast, the robot’s line speed is fixed, so the robot is encouraged to reach as many destinations as possible in a short time. In summary, the reward function serves the robot to complete the path planning efficiently.

Combining the objective’s distance and angle components, we derive the auxiliary reward function: $R=r_{\theta }\cdot r_{d}$.

## 4. Results

### 4.1. Experiment Settings

To verify the effectiveness of the algorithm, while considering different application scenarios, three turtlebot3_stage_n.world from turtlebot3 open source turtlebot3_gazebo are used to verify the effectiveness of the RND3QN algorithm for obstacle avoidance and the path specification. The first is a static obstacle environment, the second is a dynamic obstacle environment, and the third is a more challenging and complex dynamic obstacle environment. The simulation environments stage 2 and stage 3 are raster maps of size 4 × 4 and stage 4 is a raster map of size 5 × 5. As shown in Figure 7.

The simulation experiments were conducted on a laptop virtual machine configured with ubuntu 16.04, 8 GB RAM, 60 G hard drive, no graphics, and Intel(R) Core(TM) i5-11300H processor (6 processor cores were assigned to the virtual machine). Real-world experiments were conducted on a robotic turtlebot3 waffle pi. The turtlebot3 waffle pi is powered by ubuntu 18.04 and uses an NVIDIA Jetson TX2 development board with 8 GB of RAM and 32 GB of storage. It comes with its own antenna to increase the signal receiving capability. The simulated and real experiments are run at 200 episodes, 300 episodes, and 830 episodes for the three stages, and the maximum number of steps is set to 500. The hyperparameters of the algorithm are shown in the Table 3, total reward per turn and success rate are used as evaluation criteria.

### 4.2. Simulation Experiments

Training a robot using reinforcement learning in the real world is not only time consuming, but also inevitably causes damage to the robot during training. Therefore, DRL algorithms are usually trained in a simulated environment. Simulation experiments are performed in the gazebo, and to reduce the observed differences between simulation and the real world, the algorithm uses laser ranging results as abstract input. The discrete action space is not only insensitive to errors, but also easy to learn. So the discrete action space is used for the control of wheeled robots. In this experiment, the environment was reset if the robot encountered an obstacle or timed out. If the robot reaches the target location, the target location is regenerated at another location and the robot continues the task. The total reward at the end of that round may be a relatively large positive value, while the total reward at the beginning of the next round reverts to 0. Also, due to the highly random nature of the reinforcement learning algorithm, the total reward at the end of the next round may be a small positive value. So the total reward value can produce relatively large fluctuations. The results of the simulation experiments are shown in Figure 8, Figure 9, Figure 10, Figure 11, Figure 12, Figure 13 and Figure 14. Videos of the experiments can be found at https://youtu.be/guBYRJDHx1A (accessed on 9 December 2022).

For Q-learning related algorithms, the value of the value function is the predicted expectation of future rewards. If the average value function output for each time step of each episode keeps rising with training iterations, it represents that the agent learns a good strategy.

To further compare the performance of different algorithms, we define the average number of successful arrivals *p*. It reflects the maximum number of times the robot reaches the target location on average per episode.
$$\begin{array}{c} p=\frac{\sum_{i=m}^{n}goal_{i}}{n-m} \end{array}$$

DQN’ represents DQN without auxiliary reward functions and RND3QN stands for joining the ND3QN reward-based ϵ-greedy exploration strategy. The average number of successful arrivals is calculated by randomly selecting n−m rounds after the network converges, where $goal_{i}$ is the number of times the *i*-th episode reaches the target location. From the data in the Table 4, it is concluded that the success planning rate of the RND3QN algorithm improves 745%, 313% and 610% over DQN, and 173%, 65% and 60.5% over D3QN in the three simulation environments, respectively.

### 4.3. Real-World Experiments

To test the effectiveness of the algorithm in practice, we deployed the algorithm on a real robot turtlebot3 waffle pi for experiments. The robot is shown in Figure 15. The weights of 200 episodes trained in the simulated environment were assigned to the real world for path planning. The robot needs to bypass obstacles to reach the target location. Equipped with 2D LiDAR A2, turtlebot3 waffle pi robot has a detection range of 360 degrees around the robot, and the algorithm uses 24 radar detection lines.

For the real-world experiments, we laid out a map similar to turtlebot3_stage_2, as shown in Figure 16. The RND3QN algorithm was deployed to waffle pi, and the randomly generated target location was modified to generate target locations by posting topics. The target location data is of PoseStamped type. The weights of 200 episodes trained in the simulation environment are loaded for testing. The experimental results show that the robot can perform the path planning task well.

## 5. Discussion

We conducted ablation experiments on the RND3QN algorithm. The effects of double DQN, dueling DQN, n-step bootstrapping and adding reward-based exploration strategies on the performance of the algorithm are observed, respectively. The average total reward per round can effectively reflect the performance of the algorithm. The experiments show that adding the n-step approach and the reward-based exploration strategy has the greatest improvement in the algorithm performance. Finally, the effectiveness of the algorithm is demonstrated on the real robot waffle pi.

Different combinations of hyperparameters were tested in the hyperparameter selection experiments, where the number of layers of the neural network is also a very important aspect. In conjunction with the problem solved in this work, RND3QN was set up with 3 layers of size 64 fully connected layers. Two sets of optimizer combinations, RMSprop+Huber Loss and Adam+Mse Loss, were selected for comparison experiments, and the experimental results show that the latter is more effective than the former. A random gradient descent approach was chosen for network training, and the size of each batch of data was the same as the network size.

We did not solve all the problems of the DQN algorithm. Due to the small dimensionality of the state space and the dense rewards, this work does not improve the replay buffer part, which is left as future work. The traditional exploration strategy is difficult to encounter trajectories with high reward values during exploration, and the value-based approach may not pay sufficient attention to the near-optimal trajectories, resulting in slow learning or failure. To solve the problem of exploration, imitation learning, intrinsic reward, and hierarchical reinforcement learning can be introduced.

## 6. Conclusions

In this work, the RND3QN algorithm can handle local path planning and obstacle avoidance in complex unknown dynamic environments. The robot acquires data through LiDAR and passes the data through special processing as state values to the algorithm. These real-time data allow the robot to perform flexible path planning in unknown dynamic environments. At the same time, we shape the auxiliary reward function, which effectively solves the problem of non-convergence of the algorithm due to the reward sparsity of the state space. Secondly, we improve the exploration-utilization strategy to make the robot more intelligent in selecting actions. Simulation and real experimental results show that our algorithm has a higher average to reward than other algorithms for local path planning in unknown dynamic environments, and the success rate of reaching the target location is significantly improved. However, our algorithm is only applicable to discrete action spaces. A large number of actions in the continuous action space corresponds to outputting a large number of Q-values, causing memory overload. Since RND3QN uses an n-step approach, it requires more transitions, which needs more memory requirements for the algorithm, so it is necessary to adopt a more reasonable sampling method to reduce memory space. Future work will be carried out in the following areas.


Agents need more internal drivers to explore in sparsely rewarded environments. One possible solution is to introduce curiosity as an internal driver. This would increase the agent’s ability to learn, allowing for more exploratory strategic improvements.
We will explore path planning and obstacle avoidance solutions for robots in continuous action space.
A better strategy will be designed to optimize the robot’s path trajectory.
For the sample efficiency aspect, the uncertainty weights of each action will be considered. More rational and efficient sampling methods to reduce memory overhead and improve sampling efficiency will be adopted.

## Figures and Tables

**Figure 1 sensors-23-02036-f001:**
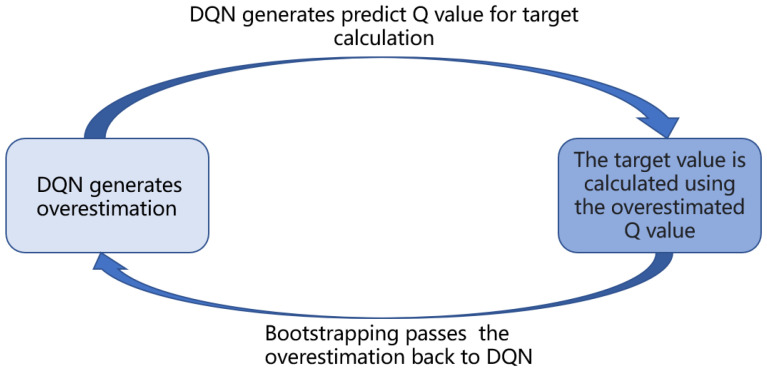
Overestimation of the process of transmission and facilitation between DQN and Target.

**Figure 2 sensors-23-02036-f002:**
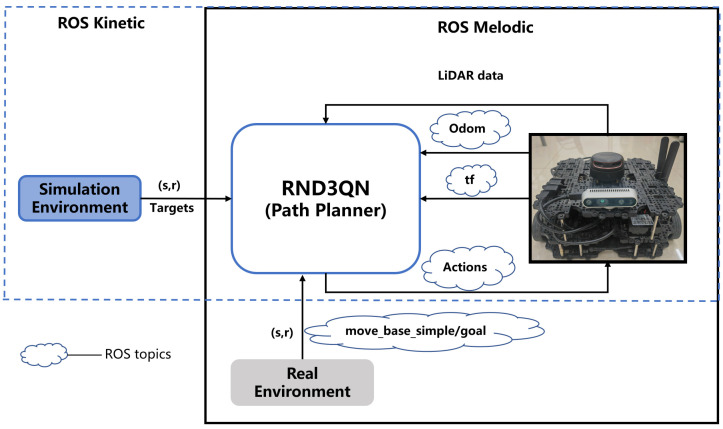
Training framework diagram for experiments based on ROS framework in simulated and real environment under different maps, respectively. The above figure depicts the planning process of the robot in the ROS framework. Firstly, the robot collects the environment data through sensors, after that the data is analyzed and passed to the agent (RND3QN), and finally the agent outputs the corresponding action instructions to be passed to the robot for execution.

**Figure 3 sensors-23-02036-f003:**
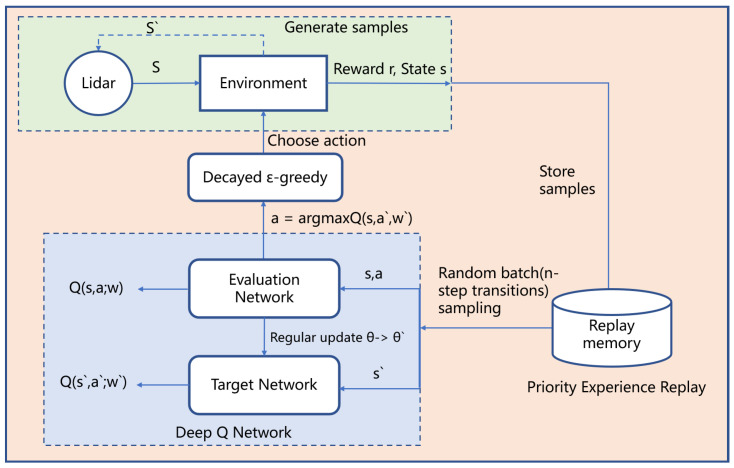
The model of the RND3QN algorithm.

**Figure 4 sensors-23-02036-f004:**
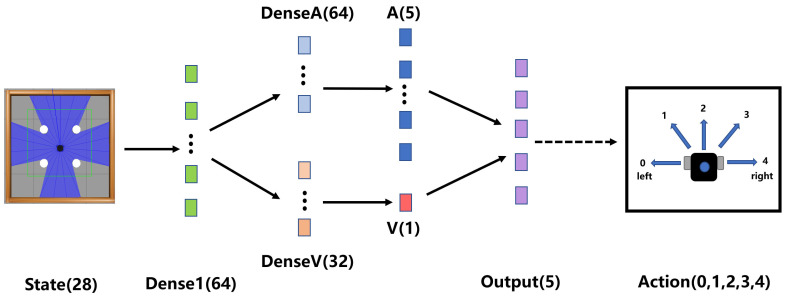
Network architecture diagram of RND3QN. The input layer is a state of size 28 dimensions and enters the branching fully connected layer after a fully connected layer of dimension size 64. The two branching fully connected layers of dimension size 64 and 32, DenseA and DenseV, are the advantage network and value network, respectively. The outputs of the two branching fully connected layers are calculated by Equation (Equation 13) to output the final Q value, and the action with the largest Q value is selected for execution.

**Figure 5 sensors-23-02036-f005:**
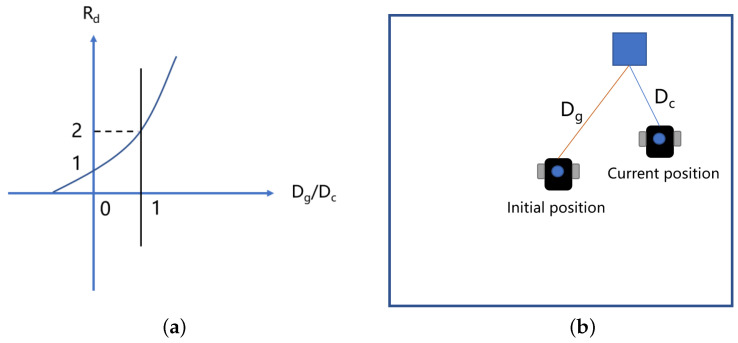
(**a**) Describes absolute distance as a function of current distance ratio and $R_{d}$ reward. (**b**) Explains absolute and relative distances.

**Figure 6 sensors-23-02036-f006:**
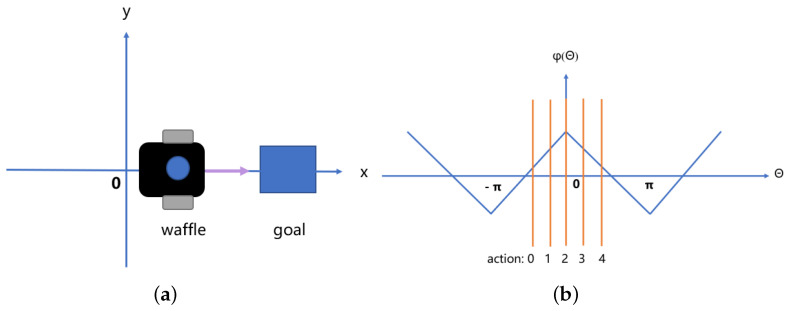
This is a graph of the function between angle and action. (**a**) Describes at this point θ=0, indicating that the robot is moving in the direction of the target. (**b**) Denotes the function of theta generated by the robot action corresponding to $\phi \left(\theta \right)$.

**Figure 7 sensors-23-02036-f007:**
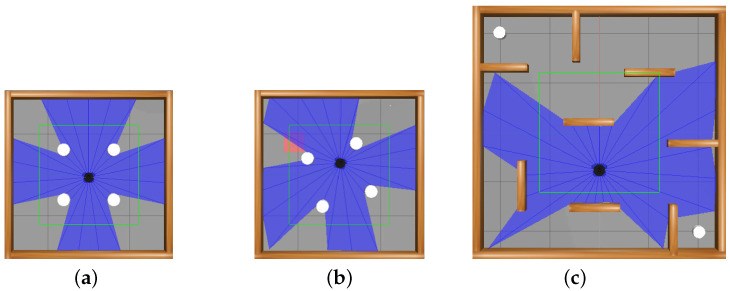
Simulation environment. (**a**) stage 2; (**b**) stage 3; (**c**) stage 4.

**Figure 8 sensors-23-02036-f008:**
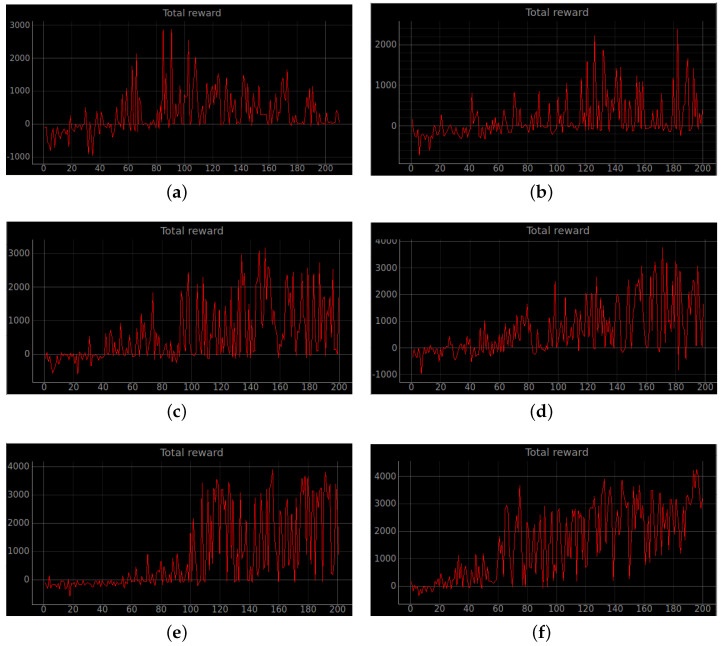
Robot path planning simulation results. The image above shows the total rewards obtained by the robot in six algorithms under stage 2, which contains an unknown environment of static obstacles. (**a**) Original DQN; (**b**) Double DQN; (**c**) Dueling DQN; (**d**) D3QN; (**e**) ND3QN; (**f**) RND3QN.

**Figure 9 sensors-23-02036-f009:**
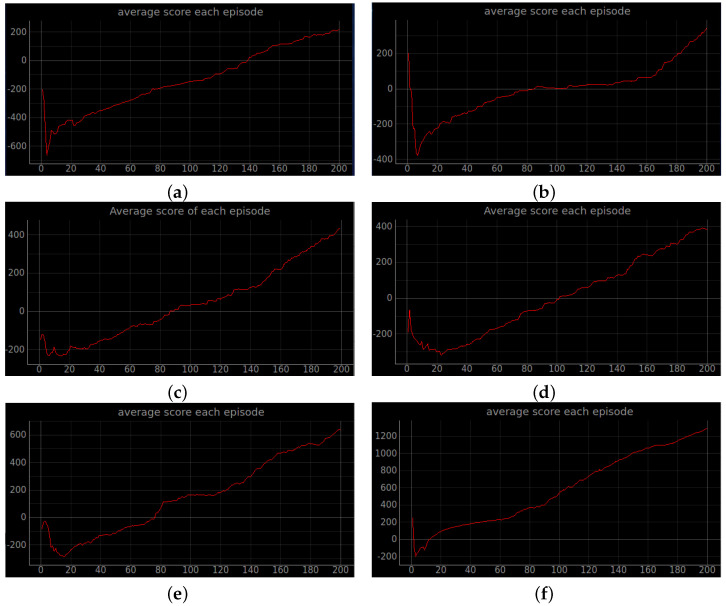
Average total reward curve per episode for stage 2. The increasing mean of the total rewards obtained from the environment per episode interaction indicates that the agent learns effective strategies. (**a**) Original DQN; (**b**) Double DQN; (**c**) Dueling DQN; (**d**) D3QN; (**e**) ND3QN; (**f**) RND3QN.

**Figure 10 sensors-23-02036-f010:**
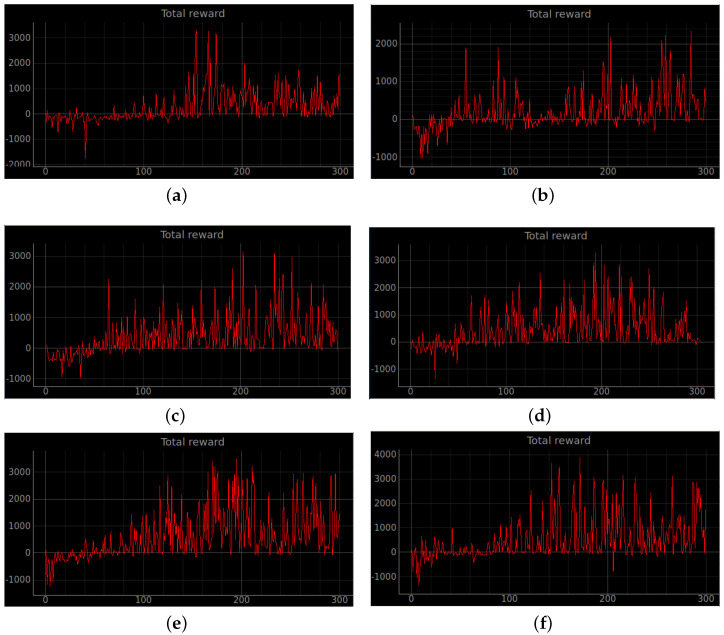
The curve of the cumulative sum of all rewards per episode for stage 3. (**a**) Original DQN; (**b**) Double DQN; (**c**) Dueling DQN; (**d**) D3QN; (**e**) ND3QN; (**f**) RND3QN.

**Figure 11 sensors-23-02036-f011:**
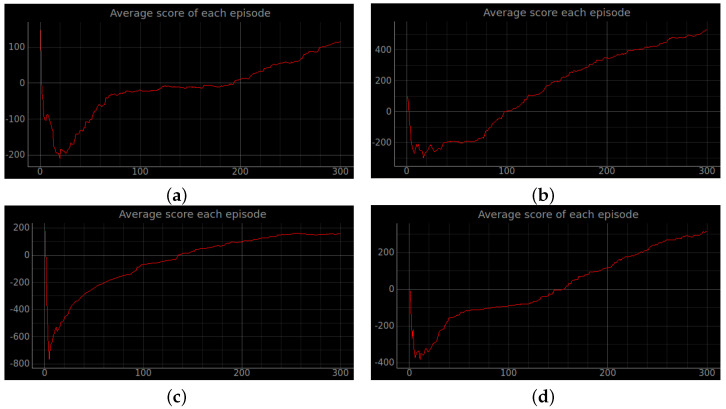
Average total reward value per episode for stage 3. The simulation results show that the episodes of RND3QN algorithm to learn an effective policy are half of those of DQN algorithm. (**a**) Original DQN; (**b**) Double DQN; (**c**) Dueling DQN; (**d**) D3QN; (**e**) ND3QN; (**f**) RND3QN.

**Figure 12 sensors-23-02036-f012:**
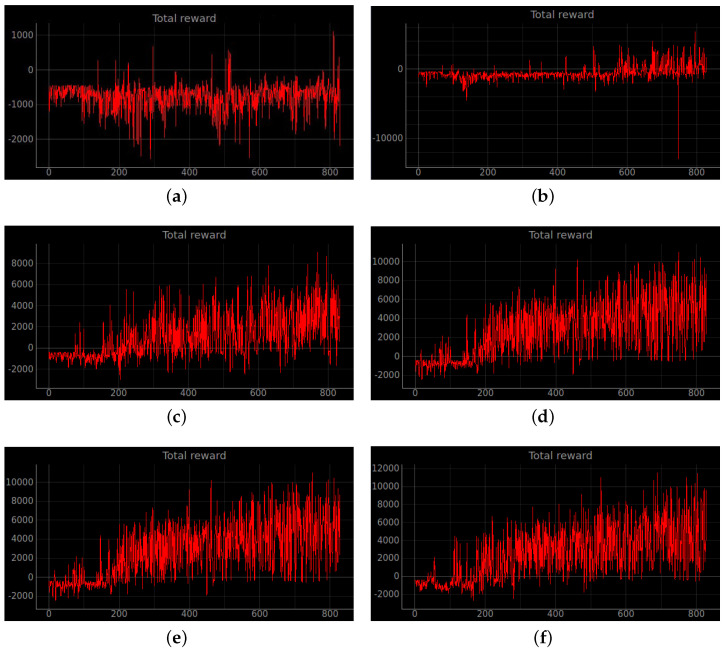
The curve of the cumulative sum of all rewards per episode for stage 4. (**a**) Original DQN; (**b**) Double DQN; (**c**) Dueling DQN; (**d**) D3QN; (**e**) ND3QN; (**f**) RND3QN.

**Figure 13 sensors-23-02036-f013:**
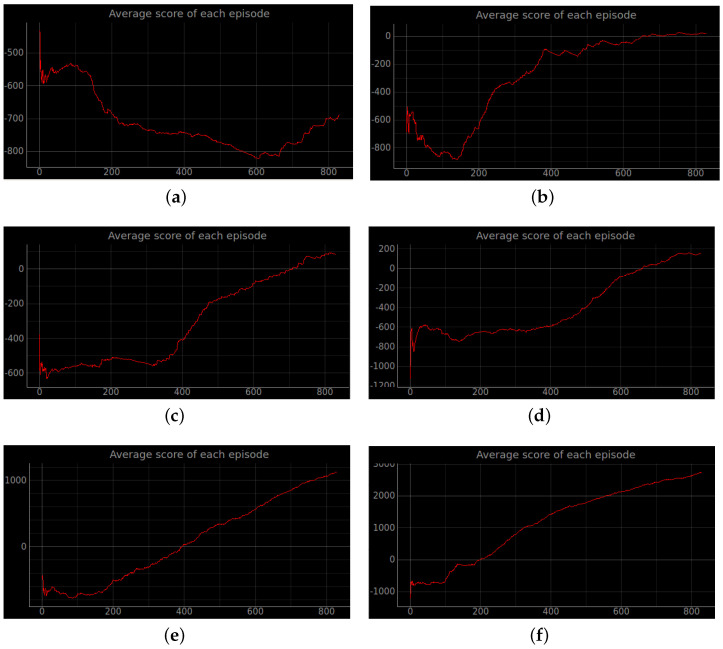
The curve of the cumulative sum of all rewards per turn for stage 4. The simulation results illustrate that the more challenging the environment is the higher the performance improvement of RND3QN compared to the DQN algorithm. (**a**) Original DQN; (**b**) Double DQN; (**c**) Dueling DQN; (**d**) D3QN; (**e**) ND3QN; (**f**) RND3QN.

**Figure 14 sensors-23-02036-f014:**
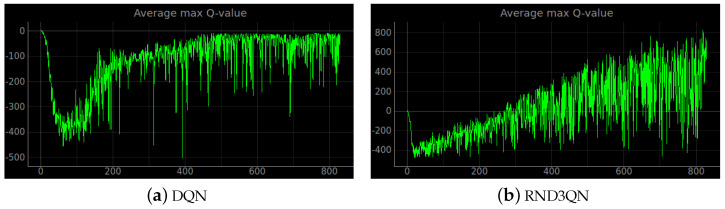
The average of the value functions for each episode in the stage 4. (**a**) Is the average curve of the maximum Q-values of the original DQN algorithm. (**b**) Is the average curve of the maximum Q-values of the RND3QN algorithm. The goal of reinforcement learning is to continuously improve the value of the value function. It can be seen from the figure that the maximum Q-values of the original DQN are all less than 0. The agent has not learned an effective policy. While the maximum Q-value of the RND3QN algorithm steadily increases.

**Figure 15 sensors-23-02036-f015:**
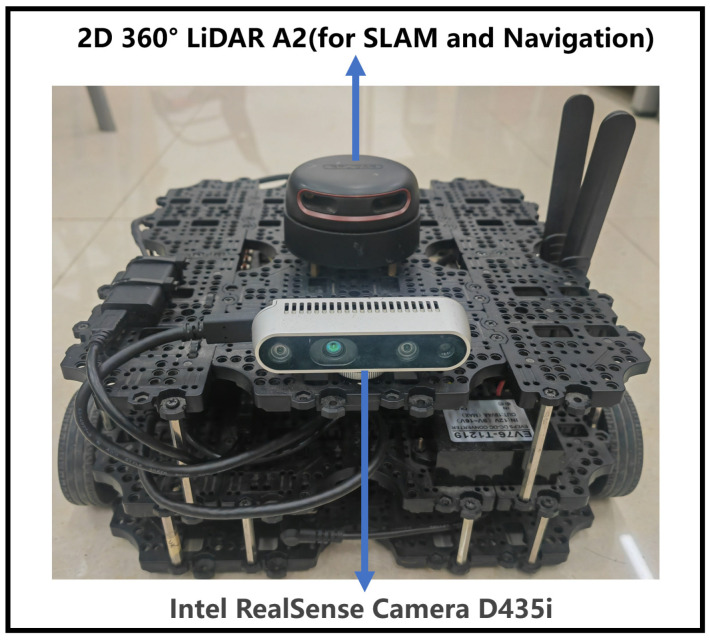
Turtlebot3 waffle pi.

**Figure 16 sensors-23-02036-f016:**
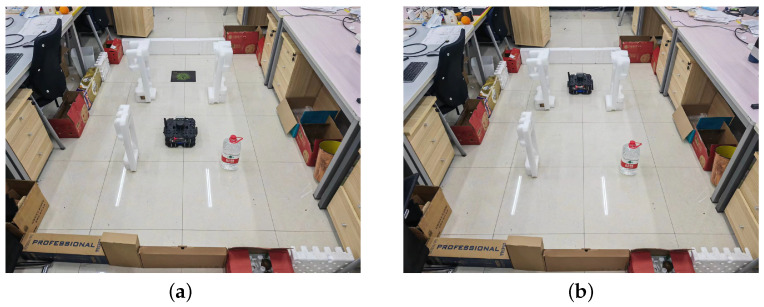
Robot performs path planning in real environments. (**a**) Initial Environment. (**b**) Robot arrives at target location.

**Table 1 sensors-23-02036-t001:** Table of action space and state space size settings in MDP. There are five actions, so the action space is 5. The sum of all dimensions of the state is 28, so the state space size is 28.

Action Space	State Space
5	28

**Table 2 sensors-23-02036-t002:** Correspondence table between motion and angular velocity.

Action	Angular Velocity (rad/s)	Direction
0	−1.5	left
1	−0.75	left front
2	0	front
3	0.75	right front
4	1.5	right

**Table 3 sensors-23-02036-t003:** Hyperparameter table.

Parameters	Value	Description
ϵ	1.0	The probability of randomly selecting an action in the ϵ-greedy strategy.
$\epsilon_{init}$	0.99	Initial ϵ (not change).
$\epsilon_{min}$	0.1	The minimum of ϵ.
learning_rate	0.00025	Learning rate.
episode_step	6000	The time step of one episode.
discount_factor	0.99	Discount factor, indicating the extent to which future events lose their value compared to the moment.
batch_size	64	Size of a group of training samples.
memory_size	1,000,000	The size of replay memory.
train_start	64	Start training if the replay memory size is greater than 64.
n_multi_step	3	Number of n-step steps.

**Table 4 sensors-23-02036-t004:** Average number of successful arrivals.

Models	DQN’	DQN	Double DQN	Dueling DQN	D3QN	ND3QN	RND3QN
stage 2	0	1.1	2.7	3.3	3.4	7.8	**9.3**
stage 3	0	0.8	0.9	1.2	2.0	2.8	**3.3**
stage 4	0	0.1	0.1	3.4	3.8	5.7	**6.1**

## Data Availability

Not applicable.

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
