# Peer review of "A Mapless Local Path Planning Approach Using Deep Reinforcement Learning Framework"

_sensors, 2023, doi:10.3390/s23042036_

Round 1

Reviewer 1 Report

This paper proposes an end-to-end local path planner based on deep reinforcement learning for unknown dynamic environments. The framework takes LiDAR data as input and outputs discrete actions using RND3QN, a proposed Deep double Q network variant. The gazebo is used for simulation and learning the policies. The experiments are also performed on the TurtleBot 3 Waffle Pi robot by transferring learned strategies in the simulation to the real robot. 

The paper, in general, is well-organized and well-written. The simulation and real-world experiments have been presented to validate the methodology. The contributions and limitations need to be concretely discussed. 

The paper could give more insights if addressing the following limitations: 

Note: Lxx means Line number xx in the paper.

  1. There are various abbreviations in the paper. Please make sure that all of them are introduced in their first usage. e.g., RND3QN (L4), DQN (L7), SLAM (L31), POMDP (L227), etc.

  2. Consistency when referencing equations is required. There are inconsistencies such as equation1 (L64), equation2 (L66), equation 3 (L243), Equation 1 (L247), Eq14 (L359), etc.  

  3. The paragraph starting from L68 needs to be clarified. The first letter in the sentence is not in the capital.

  4. The notations used in equations 1 & 2, and paragraphs starting from L71 & L75, such as r, s, a, w, Q, x, \gamma, etc., must be explained. Same with other equations too. 

  5. Please cite double DQN in L93 and dueling DQN in L94.

  6. The change of epsilon probability ... in L100 needs to be clarified. Please introduce it first. The term epsilon has been used numerous times in the paper. Is it the same as \epsilon?

  7. Please correct issues related to a consistent style. Capitalization issues such as double DQN in L106, Dueling DQN in L107, reward Return in L235, Discounted return in L242, its target (L247), in Table 1, Dueling Double DQN (L361), and so on, must be fixed. The full stops need to be included in the captions of many figures. 

  8. The contribution needs to be presented more concretely.

  9. Please clarify L126-L128. Is it correct to categorize all methods based on learning into the deep reinforcement learning category?

  10. In Sect. 2 Related work, the authors must maintain consistency in the citations. Furthermore, the reference needs to be included in L166.

  11. Please clarify the Euclidean distance in L385.

  12. In L394, you probable mean |.| not [.].

  13. The experiments are promising. For clarity, it is better to have a video and data of the experiments to validate the results further. Authors are encouraged to provide the details of the models, data, and experimental videos in the online repository.

  14. The consistency in the reference needs to be fixed. 

  15. The English have to be improved at various places in the paper. In the present form, the paper contains typos. 

Author Response

Dear Editors and Reviewers:

Thank you for the detailed comments and constructive suggestions concerning our manuscript entitleed “A Mapless Local Path Planning Approach Using Deep Reinforcement Learning Framework” for publication in Sensors.

We have revised our manuscript carefully according to the reviewers’ comments and suggestions. The detailed responses to each reviewer are listed below. Moreover, major changes of the manuscript are emphasized in red, we can remove the color in the final version.

#Reviewer 1

Note: Lxx means Line number xx in the paper.

 Comment 1: There are various abbreviations in the paper. Please make sure that all of them are introduced in their first usage. e.g., RND3QN (L4), DQN (L7), SLAM (L31), POMDP (L227), etc.

Response 1:  We thank the reviewer for providing valuable feedback. According to reviewer’s comment, we have added the full name of the acronym. See L4, L8, L13, L32, L67, L111, L158, L174, L244, L289  of the new manuscript for details.

Comment 2:  Consistency when referencing equations is required. There are inconsistencies such as equation1 (L64), equation2 (L66), equation 3 (L243), Equation 1 (L247), Eq14 (L359), etc. .

Response 2: We would like to thank the reviewer for pointing out this issue. As suggested by the reviewer, we have rewritten “equation x”  and “Equation x”to “Eq x”. See L67, L69, L71, L74, L75, L260, L264, L311, L339, L344, L345, L348, L392  of the new manuscript for details.

Comment 3:  The paragraph starting from L68 needs to be clarified. The first letter in the sentence is not in the capital.

Response 3: We are sorry that this part was not clear in the original manuscript. We have rewritten this paragraph and corrected the case of the first letter. Please See L67~L70 of the new manuscript for details.

Comment 4: The notations used in equations 1 & 2, and paragraphs starting from L71 & L75, such as r, s, a, w, Q, x, \gamma, etc., must be explained. Same with other equations too.

Response 4:  We sincerely appreciate the significant suggestions. We have added explanations to all the symbols used in the equations. Equation 1 and Equation 2: where $y_t$ in Eq 1 is the TD target value at time step t, $r_t$ is the obtained reward, $\gamma$ is the discount factor, $\underset{a}{\max Q}(s_{t+1},a;w)$ is the highest Q-value among all actions at next state $s_{t+1}$, $s_{t+1}$ is the state at time step t+1, $a$ is the action at time step t, and $w$ is the weight of the neural network. The $\alpha$ in \tEq \ref{eq2} is the learning rate, $Q(s_t,a_t;w)$ is estimated value, $y_t$ is true value. The network weights are updated by Eq 2, which makes the network converge toward the global optimal solution.

Comment 5: Please cite double DQN in L93 and dueling DQN in L94.

Response 5: We thank reviewer for reminding us this important point. We have already added these references.

Comment 6:  The change of epsilon probability ... in L100 needs to be clarified. Please introduce it first. The term epsilon has been used numerous times in the paper. Is it the same as \epsilon?

Response 6: We are sorry that this part was not clear in the original manuscript. We have introduced the ε-greedy strategy and epsilon probability. Please See L52~L56 of the new manuscript for details. Moreover, in this article, epsilon is the same as epsilon.

Comment 7: Please correct issues related to a consistent style. Capitalization issues such as double DQN in L106, Dueling DQN in L107, reward Return in L235, Discounted return in L242, its target (L247), in Table 1, Dueling Double DQN (L361), and so on, must be fixed. The full stops need to be included in the captions of many figures.

Response 7:  We are very grateful for reviewer’s careful reading. We have made careful modifications to the original manuscript,  and correct issues related to a consistent style. Please See L112, L113, L181, L186, L252, L258,  L264, L356 and so on of the new manuscript for details.

Comment 8: The contribution needs to be presented more concretely.

Response 8: We would like to thank the reviewer for pointing out this issue. In the new manuscript, we have added more detail to the contribution. Please See L117~L129 of the new manuscript for details.

Comment 9: Please clarify L126-L128. Is it correct to categorize all methods based on learning into the deep reinforcement learning category?

Response 9: We are sorry that this part was not clear in the original manuscript. The learning-based path method not only has the category of deep reinforcement learning, but deep reinforcement learning is the mainstream method. We have re-written that sentence, see L138~140 for details.

Comment 10: In Sect. 2 Related work, the authors must maintain consistency in the citations. Furthermore, the reference needs to be included in L166.

Response 10: We thank reviewer for the valuable suggestions. We have checked the to the literature carefully in Sect. 2 Related work. Note: Double DQN and double Q-learning are not from one article. So we cited different references. In addition, we have added relevant references to the L180 of the new manuscript.

Comment 11: Please clarify the Euclidean distance in L385.

Response 11: We thank the reviewer for providing valuable feedback. According to reviewer’s comment, we have added interpretation about euclidean distance to further strengthen our work. Please See L379~L382 of the new manuscript for details.

Comment 12: In L394, you probable mean |.| not [.].

Response 12: We are sorry that this part was not clear in the original manuscript. We wanted to express the effect of rounding down, but we used the wrong symbol. We have made corrections in the new manuscript, see L393.

Comment 13: The experiments are promising. For clarity, it is better to have a video and data of the experiments to validate the results further. Authors are encouraged to provide the details of the models, data, and experimental videos in the online repository.

Response 13: We sincerely appreciate the significant suggestions. We have recorded a video of the simulation experiment. Videos of the experiments can be found at https://youtu.be/guBYRJDHx1A.

Comment 14: The consistency in the reference needs to be fixed. 

Response 14: We are really sorry for our careless mistakes. We have made careful modifications to the original manuscript to ensure consistency in style throughout the text.

Comment 15:  The English have to be improved at various places in the paper. In the present form, the paper contains typos. 

Response 15: We are very grateful for reviewer’s careful reading. The typo has been corrected in the revised manuscript. Thanks for your reminder.

Again, thank you for giving us the opportunity to strength our manuscript with your valuable comments and queries. We have worked hard to incorporate your feedback and hope that these revisions persuade you to accept our submission.

Thank you and best regards.

Yours sincerely,

Yan Yin

Name: Jianwei Guo

Reviewer 2 Report

Dear Authors,

I have reviewed the attached file and have some considerations. While the proposed article is interesting, I believe that the text needs improvement. The English should be reviewed for grammar and clarity, and the authors should provide more detailed explanations on how the simulations and practical experiments were designed and implemented. Additionally, there are formatting issues with the use of uppercase and lowercase letters and some abbreviations lack clear explanations. I also suggest paying special attention to the references and including the DOI numbers.

Best regards,

Author Response

Dear Editors and Reviewers:

Thank you for the detailed comments and constructive suggestions concerning our manuscript entitleed “A Mapless Local Path Planning Approach Using Deep Reinforcement Learning Framework” for publication in Sensors.

We have revised our manuscript carefully according to the reviewers’ comments and suggestions. The detailed responses to each reviewer are listed below. Moreover, major changes of the manuscript are emphasized in red, we can remove the color in the final version.

#Reviewer 2

Note: Lxx means Line number xx in the paper.

 Comment 1: There are various abbreviations in the paper. Please make sure that all of them are introduced in their first usage.

Response 1: We thank the reviewer for providing valuable feedback. According to reviewer’s comment, we have added the full name of the acronym. See L4, L8, L13, L32, L67, L111, L158, L174, L244, L289 and L300  of the new manuscript for details.

Comment 2:  improve text writing

Response 2:  We are sorry that this part was not clear in the original manuscript. We have rewritten these paragraphs to further understand. For specific changes, please see L58~L59, L111~L116, L149~152, L294~L299, L319, L362~L363, L365~L368, L411~L412 of the new manuscript.

Comment 3:  8 and 9 are not references for this statement. These articles do not mention this stochastic approach.

Response 3: Thanks for reviewer’s significant reminding. The old manuscript The references 8 and 9 in old manuscript are not relevant to the methods described above. We have corrected the references. Take a look at L59 in the newcomer.

Comment 4: Consistency when referencing equations is required.

Response 4: We would like to thank the reviewer for pointing out this issue. As suggested by the reviewer, we have rewritten “equation x”  and “Equation x”to “Eq x”. See L67, L69, L71, L74, L75, L260, L264, L311, L339, L344, L345, L348, L392  of the new manuscript for details.

Comment 5: Missing spacing between word and number.

Response 5: We are very grateful for reviewer’s careful reading. We have added spaces between all words and numbers, such as L67, 69, etc. in the new manuscript .

Comment 6:  L68~L70: The paragraph is missing part of the text, which affecting the understanding of the text. And correlate the two pieces of text highlighted to improve context. L68: Use capital letters at the beginning of the paragraph.

Response 6: We are sorry that this part was not clear in the original manuscript. As suggested by the reviewer, we have rewritten this part. We are really sorry for our careless mistakes. We noticed that the first letter at the beginning of the paragraph was not capitalized and have corrected this issue. For details, please see L67~L70 of the new manuscript.

Comment 7: Explain the equations better and describe what each of the variables means.

Response 7: We sincerely appreciate the significant suggestions. We have added explanations to all the symbols used in the equations. For details, please see L71~L76 of the new manuscript.

Comment 8: L72: THE VARIABLES MENTIONED ARE NOT IN THE EQUATION.

Response 8:  We are very grateful for reviewer’s careful reading.  The variables here are hypothetical and are intended to further explain the overestimation problem, not the explanation of the equation above.

Comment 9: Two sentences repeated in the same paragraph, and “The” should be lower case in L73.

Response 9: We are really sorry for our careless mistakes. We thank reviewer for reminding us this important point. In L79 of the new manuscript, we have removed duplicate sentences and changed inappropriate capitalization to lowercase. For details, please see L79~L80 of the new manuscript.

Comment 10: L77: This equation should be numbered and better explained. How does this equation relate to equation 1?

Response 10: Thanks for reviewer’s significant reminding. This equation is the same as equation 1, so we have removed this equation in the new manuscript.

Comment 11: L91~L96: Very confusing paragraph, improve writing

Response 11: We are sorry that this part was not clear in the original manuscript. We have rewritten this paragraph in a new manuscript and hope it will help to understand this article more clearly. For details, please see L98~L103 of the new manuscript.

Comment 12: Find a more appropriate term and the words is inappropriate.

Response 12: Thanks for reviewer’s significant reminding. We solve this problem by either rewriting sentences or replacing vocabulary in new manuscripts to promote understanding of the text. For details, please see L95~L103, L109~L110, L111~L116, L137, L178, L253 and so on of the new manuscript.

Comment 13: Which algorithm?

Response 13: We are sorry that this part was not clear in the original manuscript. The algorithmic RDN3QN algorithm mentioned here has been pointed out in a new manuscript. For details, please see L114 of the new manuscript.

Comment 14: Figure 2 is not cited or explained anywhere in the text and terms and abbreviations used in Figure 2 are not explained in the text.

Response 14: We are very grateful for reviewer’s careful reading. We have  added the interpretation of Figure 2 in the new manuscript. For details, please see Figure 2 of the new manuscript. Thanks to the reviewers for the reminder.

Comment 15:  L151~L153: This sentence sounds redundant.

Response 15: We are very grateful for reviewer’s careful reading. After introducing the shortcomings of the RRT algorithm, we need to introduce methods for improving in this direction. Therefore, this sentence needs to be retained.

Comment 16: L267: Spelling error

Response 16: We would like to thank the reviewer for pointing out this issue. As suggested by the reviewer, we have corrected the “Distacne”, “Obsatacle” into “Distance” and “Obstacle”. For details, please see L283 of the new manuscript.

Comment 17: Table 2 was inserted without any context or explanation

Response 17: We are sorry that this part was not clear in the original manuscript. We have added explanations at Table 2 to facilitate understanding of the article. For details, please see Table 2 of the new manuscript.

Comment 18: The lines in Figure 3 are misaligned.

Response 18: We are very grateful for reviewer’s careful reading. We have revised Figure 3. For details, please see Figure 3 of the new manuscript.

Comment 19: L291: This sentence makes no sense and too repetitive.

Response 19: We are sorry that this part was not clear in the original manuscript. In order to make the article clear, we have rewritten this sentence to remove redundant parts. For details, please see L306 of the new manuscript.

Comment 20: L296: capital letter ?? or The?

Response 20: We are really sorry for our careless mistakes. We have fixed this error in L311 of the new manuscript.

Comment 21: Excessive space above equation.

Response 21: We are very grateful for reviewer’s careful reading. All excessive  space above all equations in the full text are removed.

Comment 22: Figure 4 was not well explained in the body of the text.

Response 22: We would like to thank the reviewer for pointing out this issue. In the starter draft, we added explanations at the bottom of Figure 4. For details, please see Figure 4 of the new manuscript.

Comment 23: The Figure 14 results obtained could be better explained.

Response 23: We would like to thank the reviewer for pointing out this issue. In the starter draft, we explain Figure 14 and point out the role of this evaluation indicator. For details, please see Figure 14 of the new manuscript.

Comment 24: Need to specify which table this information refers to

Response 24: We are sorry that this part was not clear in the original manuscript. We have specified specific tables in the new manuscript, and we're sorry for confusing reviewers. For details, please see L479 of the new manuscript.

Comment 25: Improper use of capitalization.

Response 25: We are very grateful for reviewer’s careful reading. We have made careful modifications to the original manuscript,  and correct the issue related to a consistent style.

Comment 26: Review the format of the references and insert the DOI number.

Response 26: We thank the reviewer for providing valuable feedback. In the new manuscript we have added the doi number after each reference except for those references without doi numbers.

Comment 27: Conclusions section, Lines 510-517. Four areas for future work are listed, but they are listed in the past tense, as if they had already been done. Since this is proposed future work, it should use future tense.

Response 27: We are sorry for the grammatical problems and have correct them based on your suggestions. Thank you very much for your comments. We have changed the past tense of future work to future tense in the new manuscript. Please see L535~L543 for details of the changes.

Again, thank you for giving us the opportunity to strength our manuscript with your valuable comments and queries. We have worked hard to incorporate your feedback and hope that these revisions persuade you to accept our submission.

Thank you and best regards.

Yours sincerely,

Yan Yin

Name: Jianwei Guo

Reviewer 3 Report

This paper presents a new method for developing path-planning using deep reinforcement learning. The main contribution of the paper is to adjust the reward function to create a potential field that increases the reward as the vehicle moves closer to the goal, rather than a single discrete reward at the goal point, thus solving the sparse reward issue. The method is presented adequately in Section 3, and the results presented in Section 4, where the authors show improvements over other reinforcement learning approaches.

Most of the necessary changes and improvements to the paper are related to grammar and presentation.

1.      Line 57, the sentence “Overestimation problem” is a fragment.

2.      Lines 68-70 appear to be a repeat of the previous sentence.

3.      Line 88, the sentence beginning “Slow convergence…” is unclear and needs to be reworded.

4.      Line 164, the sentence beginning “To solve the memory explosion…” is a fragment. Should it be combined with the following sentence using a comma?

5.      Line 291, the sentence “To reduce the errors and speed up the learning of agents” is a fragment.

6.      Line 291, the sentence beginning “The n-step TD method… is a tautology and should be reworded.

7.      Following equation 9, the state variable s should be placed into an equation environment so that it is italicized to denote it is a mathematical variable. Same comment for the advantage value, “A”, following equation 13

8.      Line 422, the sentence begging “Real-world experiments…” is repeated in the following sentence. It should be removed.

9.      Figures 8-14. The black backgrounds are difficult to see. Is it possible to change to the white background?

10.   Figures 8-14. Aren’t the lines plotted in Figure 9 the averages of the signals in Figure 8? If so, they should be plotted in the same graph, with the moving average overlayed on the original signal in a different color. Same for Figures 10-11 and Figures 12-13

11.   Conclusions section, Lines 510-517. Four areas for future work are listed, but they are listed in the past tense, as if they had already been done. Since this is proposed future work, it should use future tense.

Author Response

Dear Editors and Reviewers:

Thank you for the detailed comments and constructive suggestions concerning our manuscript entitleed “A Mapless Local Path Planning Approach Using Deep Reinforcement Learning Framework” for publication in Sensors.

We have revised our manuscript carefully according to the reviewers’ comments and suggestions. The detailed responses to each reviewer are listed below. Moreover, major changes of the manuscript are emphasized in red, we can remove the color in the final version.

#Reviewer 3

Comment 1: Line 57, the sentence “Overestimation problem” is a fragment.

Response 1: We would like to thank the reviewer for pointing out this issue. In the new manuscript we have rewritten it into sentences. See L60 of the new manuscript for details.

Comment 2: Lines 68-70 appear to be a repeat of the previous sentence.

Response 2: We are very grateful for reviewer’s careful reading. Sorry for the confusion caused by our carelessness when reading. We have removed the redundant. For specific changes, please refer to L69~L70 of the new manuscript.

Comment 3:  Line 88, the sentence beginning “Slow convergence…” is unclear and needs to be reworded.

Response 3: We are sorry that this part was not clear in the original manuscript. We have rewritten this sentence in the newcomer draft to facilitate understanding of the article. For specific changes, please refer to L95~L97 of the new manuscript.

Comment 4: Line 164, the sentence beginning “To solve the memory explosion…” is a fragment. Should it be combined with the following sentence using a comma?

Response 4: We thank reviewer for the valuable suggestions. As you said, this sentence is indeed quite long. So in the newcomer draft, we have separated this sentence with a comma. For specific changes, please refer to L178~L181 of the new draft.

Comment 5: Line 291, the sentence “To reduce the errors and speed up the learning of agents” is a fragment.

Response 5: We thank the reviewer for providing valuable feedback. We have reorganized some of the advantages of the n-step approach to language expression. Specific changes are made in L306~L307 of the new manuscript.

Comment 6:  Line 291, the sentence beginning “The n-step TD method… is a tautology and should be reworded.

Response 6: We are really sorry for our careless mistakes. Thank you very much for the reminder. This is indeed a tautological repetition, and we have reworded L306~L308 in the new manuscript to introduce the n-step method.

Comment 7: Following equation 9, the state variable s should be placed into an equation environment so that it is italicized to denote it is a mathematical variable. Same comment for the advantage value, “A”, following equation 13

Response 7: We are very grateful for reviewer’s careful reading. We ignore the issue of variables in paragraphs that need to be italicized. We have checked the literature carefully and put the variables in all paragraphs into the equation environment.

Comment 8: Line 422, the sentence begging “Real-world experiments…” is repeated in the following sentence. It should be removed.

Response 8: We are very grateful for reviewer’s careful reading. We deleted the duplicate parts according to your comments. Thank you for your comments.

Comment 9: Figures 8-14. The black backgrounds are difficult to see. Is it possible to change to the white background?

Response 9: We thank the reviewer for providing valuable feedback. We have done experiments and found that the graph with blue lines on a white background is not as clear as the graph with red lines on a black background.

Comment 10: Figures 8-14. Aren’t the lines plotted in Figure 9 the averages of the signals in Figure 8? If so, they should be plotted in the same graph, with the moving average overlayed on the original signal in a different color. Same for Figures 10-11 and Figures 12-13

Response 10: We are very grateful for reviewer’s careful reading. The lines plotted in Figure 9 are not the average of the total rewards in Figure 8. This is because our experiments were conducted in two sessions per map. The first experiment was conducted with the total reward per turn as the evaluation metric. The second experiment was conducted with the average reward per round as the evaluation metric. Because of the random nature of the reinforcement learning method the results of the two experiments may be different, it is not accurate to say that the line plotted in Figure 9 is the average of the total rewards in Figure 8. The same is true for Figures 10-11 and 12-13.

Comment 11: Conclusions section, Lines 510-517. Four areas for future work are listed, but they are listed in the past tense, as if they had already been done. Since this is proposed future work, it should use future tense.

Response 11: We are sorry for the grammatical problems and have correct them based on your suggestions. Thank you very much for your comments. We have changed the past tense of future work to future tense in the new manuscript. Please see L535~L543 for details of the changes.

Again, thank you for giving us the opportunity to strength our manuscript with your valuable comments and queries. We have worked hard to incorporate your feedback and hope that these revisions persuade you to accept our submission.

Thank you and best regards.

Yours sincerely,

Yan Yin

Name: Jianwei Guo

Reviewer 4 Report

The paper describes a local path planning method based on deep reinforcement learning. Perform tests both in simulation and in a real robot and compare the results with other methods.

Both the introduction and the related works are very elaborate and contribute great information to the paper.

The results and conclusions correctly describe the work done and show metrics comparing the system with other methods.

However, there are certain aspects that can be greatly improved.

An exhaustive review of English and grammatical aspects must be carried out.

- After each acronym, in its first appearance, indicate the full name.

- Review the wording and typography.

- It must be consistent with the writing mode, uppercase, lowercase, spaces, etc.

I will indicate the lines of the paper subject to change.

- 66-67: modify the sentence. Equation with capital E.

- Review equation 1 and 2

- 75: max(Q(s,a...) style equation parts $q$, $s$

-105: Reference ROS

-107: dueling lowercase d.

-120: . instead of ;

-123 - : Introduce the structure of the paper, sections that we are going to read.

- Figure 2: add explanation

-195-200: Phrase too long

-225-243: sometimes called "Intelligence" and sometimes "Agent".

-268: Misspelling: "Distance", "Obstacle"

-298: Remove "and"

- Equation 7 and 8: Difference between Rt and rt

- Equation 10: It is not a theorem, it is an equation.

-336-341: Equation always with a capital E

-359: Inconsistency with the way of writing "equation", "Eq". etc.

-399: Eularian??

-419;428: Figure, Table, uppercase.

-429: repetition: total reward

-Table 4: Bold the best method/result

-461: 360 degrees.

Author Response

Dear Editors and Reviewers:

Thank you for the detailed comments and constructive suggestions concerning our manuscript entitleed “A Mapless Local Path Planning Approach Using Deep Reinforcement Learning Framework” for publication in Sensors.

We have revised our manuscript carefully according to the reviewers’ comments and suggestions. The detailed responses to each reviewer are listed below. Moreover, major changes of the manuscript are emphasized in red, we can remove the color in the final version.

#Reviewer 4

Comment 1: After each acronym, in its first appearance, indicate the full name.

Response 1: We thank the reviewer for providing valuable feedback. According to reviewer’s comment, we have added the full name of the acronym. See L4, L8, L13, L32, L67, L111, L158, L174, L244, L289  of the new manuscript for details.

Comment 2:  Review the wording and typography.

Response 2:  We sincerely appreciate the significant suggestions. We have made careful modifications to the original manuscript,  and correct issues related to a consistent style.  We have also corrected any poorly worded or redundant parts. We have also corrected any poor wording or redundancies. The corrected parts in the novice draft are in red color.

Comment 3:  It must be consistent with the writing mode, uppercase, lowercase, spaces, etc.

Response 3: We are very grateful for reviewer’s careful reading. We have followed your comments to keep the writing patterns of upper and lower case, spaces, etc. consistent throughout the text to help better understand the article.

Comment 4: 66-67: modify the sentence. Equation with capital E.

Response 4: We are very grateful for reviewer’s careful reading. Sorry for the confusion caused by our carelessness when reading. We have removed the redundant and change the equation to Eq. For specific changes, please refer to L69~L70 of the new manuscript.

Comment 5: Review equation 1 and 2.

Response 5: We explain the variables in Equation 1 and Equation 2 in the new manuscript. For detailed information, please see L71~L76 of the new manuscript.

Comment 6: 75:  max(Q(s,a...) style equation parts $q$, $s$

Response 6: We are very grateful for reviewer’s careful reading.  Here q is the largest of our set Q values, and s it is an element of the set S of states. If something is not right, please provide comments to further strengthen our work.

Comment 7: 105: Reference ROS

Response 7: ROS is not a reference document; it is a robotic operating system and therefore does not need to be cited. We have added it in L111 of the new manuscript.

Comment 8: 107: dueling lowercase d.

Response 8: We are really sorry for our careless mistakes. We have corrected the capitalization issue in L113 of the new manuscript.

Comment 9: 120: . instead of ;

Response 9: We are very grateful for reviewer’s careful reading. We have changed ";" to "." in L125 of the new manuscript.

Comment 10: 123:Introduce the structure of the paper, sections that we are going to read.

Response 10: We thank the reviewer for the constructive comments. We introduce the structure of the paper and the chapters to be read in L130~L135 of the new manuscript.

Comment 11: Figure 2: add explanation.

Response 11: We are very grateful for reviewer’s careful reading. We have  added the interpretation of Figure 2 in the new manuscript. For details, please see Figure 2 of the new manuscript. Thanks to the reviewers for the reminder.

Comment 12: 195-200: Phrase too long

Response 12: We thank reviewer for reminding us this important point. In the new manuscript we have broken down this sentence to introduce the G2RL method in a shorter sentence. Please see L211~L216 of the new manuscript for details.

Comment 13: 225-243: sometimes called "Intelligence" and sometimes "Agent".

Response 13: We are sorry that this part was not clear in the original manuscript. "Intelligence" and "Agent" should both be translated as agent in reinforcement learning. Therefore, we unify them as "agent" in the new manuscript.

Comment 14: 268: Misspelling: "Distance", "Obstacle"

Response 14: We would like to thank the reviewer for pointing out this issue. As suggested by the reviewer, we have corrected the “Distacne”, “Obsatacle” into “Distance” and “Obstacle”. For details, please see L283 of the new manuscript.

Comment 15:  298: Remove "and"

Response 15: We thank reviewer for the valuable suggestions. We removed the "and" from the new manuscript based on your comments. Thank you very much for your careful guidance.

Comment 16: Equation 7 and 8: Difference between Rt and rt

Response 16: We are sorry that this part was not clear in the original manuscript. The reward "R_t" is the same as "r_t", and in the new manuscript we have modified them to "r_t" by agreement. rt ∈R.

Comment 17: Equation 10: It is not a theorem, it is an equation.

Response 17: We thank the reviewer for providing valuable feedback. According to reviewer’s comment, we have changed Equation 10 from a theorem to an equation.

Comment 18: 336-341: Equation always with a capital E. 359: Inconsistency with the way of writing "equation", "Eq". etc.

Response 18: We would like to thank the reviewer for pointing out this issue. As suggested by the reviewer, we have rewritten “equation x”  and “Equation x”to “Eq x”. See L67, L69, L71, L74, L75, L260, L264, L311, L339, L344, L345, L348, L392  of the new manuscript for details.

Comment 19: 399: Eularian??

Response 19: We are very grateful for reviewer’s careful reading. We used the wrong term, we wanted to say "Eulerian" instead of "Eularian". We apologize for the confusion in your reading.

Comment 20: 419;428: Figure, Table, uppercase.

Response 20: We thank the reviewer for providing valuable feedback. We have standardized the formatting by changing the full-text figure and table to all caps.

Comment 21: 429: repetition: total reward

Response 21: We would like to thank the reviewer for pointing out this issue. In the new manuscript we have removed repetitions to make the reading clear.

Comment 22: Table 4: Bold the best method/result

Response 22: We sincerely appreciate the significant suggestions. We have bolded the best methods and results in Table 4 in the novice draft. Thank you for the reminder.

Comment 23: 461: 360 degrees.

Response 23: We appreciate the guidance you provided and we have revised line L487 of the new manuscript based on your comments.

Again, thank you for giving us the opportunity to strength our manuscript with your valuable comments and queries. We have worked hard to incorporate your feedback and hope that these revisions persuade you to accept our submission.

Thank you and best regards.

Yours sincerely,

Yan Yin

Name: Jianwei Guo

Round 2

Reviewer 1 Report

The authors have addressed most of my previous comments. The paper has improved a lot. It needs one more round of proofreading. The following issues need to be addressed by the authors.

Note: Lxx means Line number xx in the paper.

  1. In L56, L58, L110, L128, L299, L375, Algorithm 1, and Algorithm 2, you mean \epsilon instead of epsilon.

  2. The sentence in L70-L71, ".. causes the overestimated value to become more and more overestimated." is unclear. 

  3. In L175, MDP should not be in bold.

  4. In Table 2, you mean left front instead of Left front.

  5. Full stops at the end of the picture cation are missing in Figures 3, 15, and 16.

  6. In Figure 7, the First letter of the sentence should be capitalized.

  7. In L378, you probably want to write \epsilon instead of \varepsilon.

  8. The authors are encouraged to have real-world experiments too. 

Author Response

We would like to extend our most sincere appreciation to the reviewer's valuable comments. According to the comments, we have improved the quality of our work, and highlighted the changes with red text in the revised manuscript. Please see below for a point-by-point response to the comments.

Comment 1: In L56, L58, L110, L128, L299, L375, Algorithm 1, and Algorithm 2, you mean \epsilon instead of epsilon.

Response 1: We think this is an excellent suggestion. We mentioned \epsilon in the strategy "\epsilon-greedy" at the time of writing, but \epsilon is the same as epsilon. Thank you for your comments, which made me realize how confusing this can be. We have replaced all epsilon with \epsilon in the new manuscript.

Comment 2: The sentence in L70-L71, ".. causes the overestimated value to become more and more overestimated." is unclear. 

Response 2: We thank the reviewer for providing valuable feedback. We have rewritten the phrase L70~L71, which actually corresponds to the arrow "BootStraooing passes the overestimation back to DQN" in Figure 1.

Comment 3: In L175, MDP should not be in bold.

Response 3: In view of this valuable suggestion, we revise the content of the new manuscript. Removed bold from MDP.

Comment 4: In Table 2, you mean left front instead of Left front.

Response 4: We are very grateful for the reviewer’s careful reading. We were really sorry for our careless mistakes. Thank you for your reminder. We have changed "Left front" to "left front" in the new manuscript.

Comment 5: Full stops at the end of the picture cation are missing in Figures 3, 15, and 16.

Response 5: Thank you for the reminder. We weren't very clear about some of the formatting, and we apologize for making a lot of low-level mistakes in the text. Thank you again for your careful guidance, and we have corrected these errors in the new manuscript.

Comment 6: In Figure 7, the First letter of the sentence should be capitalized.

Response 6: We thank the reviewer for reminding us of this important point. We have corrected this error in the new manuscript.

Comment 7: In L378, you probably want to write \epsilon instead of \varepsilon.

Response 7: You are right, what we want to express is \epsilon. we have recognized the need for format uniformity and have already made changes in L378 in accordance with your comments.

Comment 8:  The authors are encouraged to have real-world experiments too. 

Response 8: We would like to thank the reviewer for pointing out this issue. We agree with your point of view. It's true that the article on real experiments was only done on stage 2, which is less informative. But since we are currently on winter break, we are not able to get a real robot to do the experiment. We hope you can understand.

Once again, thank you very much for your valuable comments and suggestions. We hope that the revisions in the manuscript and our accompanying responses will be sufficient to make our manuscript suitable for publication in Sensors.

Best regards.

Yours sincerely,

Yan Yin

Name: Jianwei Guo
